# Surgical Interventions for Inferior Turbinate Hypertrophy: A Comprehensive Review of Current Techniques and Technologies

**DOI:** 10.3390/ijerph18073441

**Published:** 2021-03-26

**Authors:** Baharudin Abdullah, Sharanjeet Singh

**Affiliations:** Department of Otorhinolaryngology-Head & Neck Surgery, School of Medical Sciences, Universiti Sains Malaysia, Kubang Kerian 16150, Kelantan, Malaysia; sharanleyl80@gmail.com

**Keywords:** inferior turbinate hypertrophy, turbinectomy, turbinoplasty, laser, cryotherapy, electrocautery, radiofrequency, microdebrider, coblation

## Abstract

Surgical treatment of the inferior turbinates is required for hypertrophic inferior turbinates refractory to medical treatments. The main goal of surgical reduction of the inferior turbinate is to relieve the obstruction while preserving the function of the turbinate. There have been a variety of surgical techniques described and performed over the years. Irrespective of the techniques and technologies employed, the surgical techniques are classified into two types, the mucosal-sparing and non-mucosal-sparing, based on the preservation of the medial mucosa of the inferior turbinates. Although effective in relieving nasal block, the non-mucosal-sparing techniques have been associated with postoperative complications such as excessive bleeding, crusting, pain, and prolonged recovery period. These complications are avoided in the mucosal-sparing approach, rendering it the preferred option. Although widely performed, there is significant confusion and detachment between current practices and their basic objectives. This conflict may be explained by misperception over the myriad of available surgical techniques and misconception of the rationale in performing the turbinate reduction. A comprehensive review of each surgical intervention is crucial to better define each procedure and improve understanding of the principle and mechanism involved.

## 1. Introduction

Chronic nasal obstruction is caused by either nasal or septal deformities as well as mucosal disease associated with turbinate hypertrophy. Turbinate hypertrophy is observed in various conditions, including allergic rhinitis, vasomotor rhinitis, and infectious rhinitis. Medical treatments such as antihistamines, topical decongestants, and topical corticosteroids are commonly used to treat those conditions, principally to reduce nasal obstruction and restore comfortable nasal breathing [1]. However, not all patients may respond to those medications. There might be slight improvement, or in some cases, there is no response at all and they end up experiencing persistent nasal block. Surgical reduction of the inferior turbinate is warranted to relieve the nasal block caused by the hypertrophied inferior turbinates. Surgical reduction of the inferior turbinate involves removal of the mucosa, soft erectile tissue, and turbinate bone. Different techniques have been applied to increase the nasal airway passage, such as conventional turbinectomy, laser turbinectomy, cryoturbinectomy, electrocautery turbinectomy, conventional turbinoplasty, microdebrider turbinoplasty, coblation turbinoplasty, radiofrequency turbinoplasty, and ultrasound turbinoplasty. Overall, the surgical techniques are classified into two types, the mucosal-sparing and non-mucosal-sparing, based on the preservation of the medial mucosa of the inferior turbinate (Table 1).

Conventional turbinectomy (total or partial) is considered very effective in relieving nasal block. Due to the excessive loss of tissue (bone and mucosa), the postoperative complications include excessive bleeding requiring blood transfusion, crusting, pain, and prolonged recovery period. Hence, a more mucosal-friendly approach is preferred; the turbinoplasty procedure, which resects either soft tissue or bone or both with preservation of the mucosa. Additionally, out-fracture of the turbinate may be performed to augment the other procedures. With the availability of the videoendoscopic system, the majority of the surgical techniques are currently being performed endoscopically. Irrespective of the techniques, the aims are to obtain relief of nasal obstruction, preserve the function of the turbinate, and avoid complications such as bleeding, crusting, and excessive pain [2]. To achieve this, adequate inferior turbinate tissue must be removed while avoiding excessive turbinate mucosa resection. In this review, we define each surgical technique to highlight the principle and mechanism involved in each procedure.

### 1.1. Surgical Anatomy of the Inferior Turbinate

The inferior nasal turbinate consists of turbinate bone, mucoperiosteum, soft erectile tissue, and mucosa [3]. The irregular turbinate bone is interspersed with blood vessels and covered by the mucoperiosteum. It is located at the inferior part of the nose extending from the anterior nostril to posterior choana. The turbinate bone is connected to the palate, ethmoid, and lacrimal sac. The inferior nasal turbinate is expansible due to the presence of the submucosal cavernous plexus, especially well developed at its anterior part. Among the triggering factors are allergy, infection, or hormonal changes, causing engorgement of the turbinates. Permanent hypertrophy occurs due to persistent and excessive stimulation.

The inferior turbinate is supplied by inferior thyroid artery (ITA), a branch of the posterior lateral nasal artery [4]. The inferior thyroid artery enters the inferior turbinate at its posterosuperior end, where it divides into two or three branches. It courses through the inferior turbinate in a bony canal wrapped by a fascial coat binding the ITA and canal tightly. This relation is the main reason for the prolonged bleeding following turbinate surgery, as the fascial coat prevents the ITA from contracting.

### 1.2. Evaluation of the Inferior Turbinate Hypertrophy for Surgical Intervention

There are several classification systems for inferior turbinate hypertrophy. Three of them are based on clinical examination of the inferior turbinate size, while one is based on computed tomography scan imaging [5,6,7,8]. To assess the outcome of submucous resection in their patients, Friedman et al. [5] used the three-grading system; grade 1 is mild enlargement with no obvious obstruction, grade 2 is incomplete obstruction, and grade 3 complete occlusion of the nasal cavity. A study correlating the nasal anatomy to the severity of obstructive sleep apnea classified inferior turbinate hypertrophy as 0—normal, 1—mild, 2—moderate, or 3—severe [6]. Camacho et al. [7] classified the inferior turbinate’s size as four grades based on its position in the total nasal airway space as visualized on nasoendoscopic assessment. When the inferior turbinate occupies 0 to 25% of total airway space, it is grade 1, grade 2 is 26% to 50%, grade 3 is 51% to 75%, and grade 4 is 76% to 100% (Figure 1). By studying the inferior turbinate bone using computed tomography, Uzun et al. [8] rated the inferior turbinate size as type 1 (lamellar), type 2 (compact bone), type 3 (combined), and type 4 (bullous). There is no consensus or standardization on the ideal classification system, and this creates difficulty and confusion in assessing effectiveness of any given technique for surgical reduction of inferior turbinate hypertrophy. We prefer the Camacho classification system [7] in our clinical practice, as it provides an objective, practical, and reliable assessment. It is imperative during the evaluation to exclude the presence of septal deviation, allergic rhinitis, or unilateral inferior turbinate hypertrophy [9]. The concomitant conditions should be addressed simultaneously with the hypertrophied turbinate to achieve the best outcome.

Nasal obstruction owing to inferior turbinate hypertrophy can be measured objectively using an active anterior rhinomanometry, by measuring the differential pressure between anterior and posterior portions of the nose [10]. It is derived from the volume of airflow in the nose, measured separately both before and after the administration of a decongestant. By this way, the amount of mucosal or bony contribution towards the nasal obstruction can be estimated. Any increase in the nasal mucosal impedance caused by the nasal obstruction can be identified by this evaluation, but it cannot localize the site of intranasal obstruction. For better accuracy, it should be applied concurrently with acoustic rhinometry. The reflection of acoustic signal by the respective nasal anatomical components forms the basis of acoustic rhinometry measurement [10]. By this means, the volume of all nasal sections can be measured, which allows a reliable topographic obstruction to be identified. Corresponding to the active anterior rhinomanometry, the measurement before and after decongestion enables it to differentiate mucosal pathology from other causes. This procedure has good reproducibility and is quick and noninvasive, which makes it advantageous for examination of nasal pathology in children. It has become the first-line method to confirm the location and level of nasal obstruction, and is universally used as an objective measurement of the effectiveness of medical and surgical treatments.

### 1.3. Indications for Inferior Turbinate Reduction Surgery

One of the most common manifestations of chronic rhinitis is nasal obstruction. Nasal obstruction occurs as a result of submucosal or mucosal hypertrophy associated with increased vascularity of the inferior turbinate. Treatment of inferior turbinate hypertrophy consists of topical intranasal corticosteroid sprays, oral antihistamines, and topical decongestants. When conservative management fails within the appropriate treatment period, surgical treatment is indicated. Generally, the consensus for the point of surgical intervention is at 3 months, ensuing failure of medical therapy to resolve nasal obstruction as a result of the turbinate hypertrophy [11]. However, if there is a concomitant rhinosinusitis, the medical treatment could be extended up to 6 months. There are two types of surgical techniques, the mucosal-sparing and non-mucosal-sparing, based on the preservation of the medial mucosa of the inferior turbinate.

## 2. Nonmucosal Preservation Surgery

### 2.1. Conventional Turbinectomy

Turbinectomy involves removal of all or a portion of the inferior turbinate and may be performed by direct visualization or with the aid of an endoscope. The degree of excision can be anywhere from limited to complete resection depending upon the degree of the hypertrophy and includes turbinate mucosa, soft erectile tissue, and bone. Usually the inferior turbinate is resected using an angled scissor along its insertion at the lateral nasal wall. Elwany et al. [12] compared partial turbinectomy against three other techniques for inferior turbinate reduction to determine their subjective and objective outcomes. Their study found both partial turbinectomy and laser turbinectomy outperformed the other two techniques of inferior turbinoplasty and cryoturbinectomy in terms of relieving nasal obstruction and improving olfaction. In addition, all four techniques did not affect mucociliary clearance. However, the authors observed those patients who underwent partial turbinectomy developed nasal discomfort, headache, atrophic changes, and postoperative bleeding more than others. In a large trial of 457 patients, Passali et al. [13] reported that inferior turbinectomy conferred significant relieve of nasal obstruction but was associated with more complications, notably, intense pain, crusting, and bleeding. Atrophic rhinitis and empty nose syndrome were recognized as late sequalae of this procedure, especially following total turbinectomy [14]. Empty nose syndrome is a disorder characterized by paradoxical nasal obstruction in the presence of a wide patent nasal cavity [14]. The excessive dryness may also lead to atrophic rhinitis with crusting and nasal block. Crusting may develop due to the disruption of mucociliary clearance, raw mucosal edges, and exposed bone. Besides cold microinstruments, turbinectomy may also be performed by laser, electrocautery, and cryosurgery.

### 2.2. Laser Turbinectomy

The lasers commonly used for inferior turbinate reduction are diode and CO_2_ lasers. Other type of lasers such as Neodymium-doped: yttrium aluminum garnet (Nd-YAG), Holmium: YAG, potassium titanyl phosphate (KTP), and argon plasma lasers have been reported [15]. The properties differ between the laser types based on their application of contact or noncontact mode, pulsed or continuous wave emission, emitted wavelength, and output power. The diode laser is the preferred choice for inferior turbinate reduction as it provides accurate cutting with adequate hemostasis. It has an infrared wavelength spectrum of 805–980 nm, apt for cutting soft tissue and suitable for endoscopic sinus surgery. It provides a coagulation effect when the setting is at pulse mode and low power output, and precise cutting effect with the ability to vaporize mucosa, cartilage, or bone, at continuous mode and high energy output. A study was performed by Sroka et al. [16] comparing the use of Holmium: YAG against diode lasers for turbinate reduction. The two techniques showed significant improvement in nasal patency over six months and three years postoperatively, with subjective improvement of nasal breathing described in 67.5% of patients following Holmium: YAG laser and 74.4% following diode laser treatment. Both showed significant improvement of nasal flow on rhinomanometry. There were no immediate complications such as major bleeding post-surgery for either the Holmium: YAG or diode laser treatment, with minor complications such as pain or slight bleeding reported in 5% to 8% of cases. A randomized clinical trial comparing diode laser with radiofrequency was conducted by Kisser et al. [17]. The patients were randomized to laser therapy in one arm, and the other arm was subjected to radiofrequency therapy. A significant reduction of nasal obstruction was observed for both laser and radiofrequency therapies after three months. There were no major complications observed, but significant discomfort was reported in the radiofrequency arm.

Prokopakis et al. [18] compared the use of CO_2_ laser with radiofrequency and electrocautery to evaluate their outcome and effect on nasal obstruction using subjective visual analogue scale and objective rhinomanometry. The study found 86% of patients in the CO_2_ laser group reported subjective improvement in nasal obstruction, but no statistical difference was found among these three groups. Long-term impairment of the mucociliary transport and a more challenging manipulation of the device represent the major disadvantages of CO_2_ laser.

### 2.3. Electrocautery Turbinectomy

This technique involves an application of electrical current to cauterize the turbinate tissue either on the mucosal surface or in a submucosal plane. It is the least effective in improving nasal airway resistance and reducing turbinate volume, with higher rates of postoperative crusting and nasal synechiae reported [13]. When performed submucosally, the amount of tissue destruction is hard to gauge and there is substantial risk of surrounding tissue destruction from the excessive temperatures generated, owing to the requisite high power and voltage. 

### 2.4. Cryoturbinectomy

Cryotherapy is a minimally invasive procedure that uses nitrous oxide or liquid nitrogen as a cooling agent and induces necrosis by freezing the turbinate. It works by inducing scarring and direct destruction of mucosa and submucosal erectile tissue. The overall short-term results are satisfactory, but the benefit is usually not sustainable. The amount of volume reduction is hard to predict, and compared to the other methods, it has dismal long-term results [13]. Cryosurgery was later gradually abandoned due to the availability of new techniques enabling better procedures to be performed. 

## 3. Mucosal Preservation Surgery

### 3.1. Conventional Turbinoplasty 

This surgery is designed to remove the nonfunctional obstructive part of the turbinate while preserving the functional medial mucosa, which plays the key role in the warming and humidification of air through the nasal passages. Performed endoscopically, inferior turbinoplasty has the advantage over the other turbinate procedures by preserving sufficient mucosa, while removing adequate obstructed tissue to improve the airway significantly. The other term used for this technique is “submucosal resection”, as a reference to its submucosal dissection procedure. There are two types of turbinoplasty: intraturbinoplasty and extraturbinoplasty [19]. An intraturbinoplasty is a technique involving tunneling inside the turbinate, which only removes the submucosal erectile tissue, leaving behind the bulky inferior turbinate bone. This procedure is meant to address inferior turbinate hypertrophy contributed by the soft erectile tissue. When both soft erectile tissue and turbinate bone are removed, it is designated as an extraturbinoplasty. The extraturbinoplasty is a modification of an inferior turbinoplasty that combines conservative sparing of the nasal mucosa together with the removal of the obstructing soft tissue and part of the bulky inferior turbinate bone. An intraturbinoplasty may be performed by microdebrider, coblation, radiofrequency, and ultrasound, whereas extraturbinoplasty may be performed by microinstruments, coblation, and microdebrider.

An extraturbinoplasty utilizing the medial mucosa as a medial flap to cover the raw edges of resected lateral mucosa and soft erectile tissue is initially started as a nonendoscopic procedure [20]. One of the observations by Mabry in his series was that over 30% of his patients failed to obtain relief from postnasal discharge and rhinorhea at one year and more postoperatively [20]. This was attributed to an excessive mucosal preservation, as without an endoscope to guide the resection, the volume of reduction was imprecise. Putterman et al. [21] later reported a modification of an extraturbinoplasty, which was performed endoscopically. This technique involved making an incision at the anterior inferior turbinate using cold instruments such as sickle knife or micro scissor. The incision was extended downwards and along the inferior surface of inferior turbinate to its posterior end. Using microinstruments (microscissor and/or cutting forceps), the whole lateral aspect of the inferior turbinate mucosa and soft tissue were removed in an anterior to posterior direction. The turbinate bone was dissected off the soft tissue using a freer dissector or microscissor to separate it from the medial mucosa of the inferior turbinate, and forceps was used to remove it (Figure 2). The resected posterior end of the inferior turbinate was cauterized to prevent postoperative bleeding. Following the removal of bone and lateral mucosa, the medial mucosa was rotated laterally to cover the remaining exposed area of the lateral part of the inferior turbinate. A modification of this technique using microdebrider was later performed by Barham et al. [22]. Employing this procedure to treat patients suffering from inferior turbinate hypertrophy attributed to allergic and nonallergic rhinitis, Hamerschmidt et al. [23] reported a marked improvement in nasal obstruction (94.7%). Besides a reduction of nasal obstruction, their patients also demonstrated improvement in snoring (89.5%), smell (100%), facial pressure (95.5%), and allergy symptoms of nasal itching, runny nose, and sneezing (89.7%).

### 3.2. Microdebrider Turbinoplasty

Following its introduction in the otorhinolaryngological specialty, the microdebrider made a significant advancement in endoscopic sinus surgery [24]. It became widely accepted as a useful tool for sinus surgery with the capability of continuously suctioning blood from the operative site, allowing better surgical visual field and permitting precise tissue removal in a mucosal-sparing manner. Barham et al. [22] described the steps of using a microdebrider for an extraturbinoplasty medial flap technique. A window was created at the anterior inferior turbinate using the microdebrider blade. Using the microdebrider blade, the whole lateral aspect of the inferior turbinate mucosa and soft erectile tissue were removed in an anterior to posterior direction (Figure 3). The turbinate bone was dissected off the soft tissue using a cottle dissector to separate it from the medial mucosa of the inferior turbinate (medial flap), and a Blakesley forceps was used to remove it. If there was any bleeding encountered, bipolar cautery was used for hemostasis. Postremoval of bone and lateral mucosa, the medial flap was placed in its final position curving inferolaterally to cover the remaining exposed area of the lateral inferior turbinate.

In a prospective, randomized, comparative trial by Joniau et al. [25], microdebrider turbinoplasty was compared against electrocautery in patients with chronic nasal obstruction secondary to inferior turbinate hypertrophy. Subjective symptoms questionnaires, endoscopic scoring, and acoustic rhinometry showed microdebrider turbinoplasty was the superior method in achieving relief of nasal obstruction. Rapid reduction of inferior turbinate size was apparent at one week postoperatively and persisted until one year. Nasal breathing was better, and no crusting was observed. The observed changes were confirmed by acoustic rhinometry. Improvement of nasal obstruction and size reduction were slower in electrocautery, attained at three weeks, with relapse at one year. Another study evaluated the objective outcome by total nasal resistance using anterior rhinomanometry in patients with substantial nasal congestion due to perennial allergic rhinitis [26]. Post microdebrider turbinoplasty, the total nasal resistance significantly decreased from 0.45 Pa/cm^3^ per second preoperatively to 0.28 Pa/cm^3^ per second at one year postoperatively. Besides the reduction of nasal resistance, their quality of life also showed significant improvement. 

As a surgical tool, the microdebrider has a distinct advantage for its versatility. Besides its application in extraturbinoplasty, it can also be used for an intraturbinoplasty or endoscopic sinus surgery for a concomitant rhinosinusitis when this is necessary. The surgical steps of an intraturbinoplasty were described by Lee et al. [27]. An intraturbinoplasty was performed by initially creating a submucosal pocket at the anterior pole of the inferior turbinate. The microdebrider was then inserted to resect and remove the submucosal erectile tissue with or without the turbinate bone removal. Both the medial and lateral mucosal surfaces were preserved. 

The application of the microdebrider as an intraturbinoplasty method has been shown to have parallel outcome to that of extraturbinoplasty technique. Lee [19] conducted a comparative study between microdebrider intraturbinoplasty and extraturbinoplasty techniques and showed that both methods exhibited significant improvement in the degree of nasal obstruction, rhinorrhea, sneezing, nasal itching, and postnasal drip 12 months postoperatively, but the intraturbinoplasty method had an edge of a shorter operative time. Another study compared microdebrider intraturbinoplasty with the conventional turbinoplasty method [28]. Both techniques were shown to significantly improve the nasal symptoms and increase volume of the nasal patency, but microdebrider intraturbinoplasty had statistically significant lower blood loss and a shorter operative time. Hence, the study concluded that it provides marked improvements in nasal obstruction and has the advantage of mucosal preservation alongside controlled volume reduction, minimal trauma, reduced bleeding, and enhanced precision.

A comparative study was conducted by Cingi et al. [29] to assess microdebrider turbinoplasty with radiofrequency turbinoplasty. Patients were assigned to the microdebrider arm and the radiofrequency arm. Symptom improvement was statistically significant in microdebrider group on the seventh day and first and third months after surgery. The rhinomanometric measurements showed wider nasal patency in the microdebrider group when compared against the radiofrequency group. However, the use of microdebrider still has risk of postoperative bleeding. A study by Lee et al. [27] found 30% of patients developed postoperative nasal bleeding, requiring temporary packing with epinephrine-soaked gauze following microdebrider intraturbinoplasty. The average intraoperative blood loss was about 10 mL, with other minor issues such as crusting, synechiea, and throat dryness. The authors compared their results to those performed in European patients and postulated that Asian and Caucasian noses have a different innate immunological mechanism. Due to this, they recommended under-resection of the intraturbinal tissue to prevent some of the complications observed.

### 3.3. Coblation Turbinoplasty

Coblation is a unique method of delivering radio frequency energy to the soft tissue for applications in otolaryngology. By using radio frequency in a bipolar mode with a conductive solution, such as saline, it energizes the ions in the saline to form a small plasma field. The decreased thermal effect consequently leads to less pain and faster recovery for cases where tissue is excised [30]. Coblation induces reduction of the inferior turbinate by vaporizing and destroying the soft erectile tissue. The volume reduction and tissue fibrosis are immediate and sustainable. Further swelling and hypertrophy of inferior turbinate are prevented by contracture and anchoring of the mucosa to the periosteum as a result of the fibrosis. Coblation technology may be performed as an intraturbinoplasty or extraturbinoplasty technique (Figure 4). Plain saline is initially injected in the turbinate before its activation and insertion. Passali et al. [31] performed a coblation intraturbinoplasty by using the Coblator II surgery system and a Reflex Ultra 45 wand set at power level four. After activation, the tip of the wand was introduced at the anterior part of the inferior turbinate to create a horizontal channel from anterior to posterior. The wand was advanced along the length of the turbinate submucosally. Depending on the bulk of the turbinate, these steps could be repeated by creating an additional one to two channels. Using this surgical technique, they compared it with radiofrequency turbinoplasty and other techniques. Both coblation and radiofrequency were found to have comparable outcomes in improving the nasal patency and relieving the nasal blockage. The outcome of both coblation and radiofrequency was better than that of conventional turbinoplasty but equivalent to that of conventional turbinectomy with minimal complications.

Di Rienzo Businco et al. [32] conducted a study to determine the efficacy of coblation tunneling technique on patients with persistent inferior turbinate hypertrophy. The procedure was done by inserting the coblation wand tip through the anterior head of inferior turbinate all the way posteriorly in a submucosal compartment. Postoperatively, both subjective symptoms score and objective rhinomanometry evaluation showed significant improvement in their symptoms and apparent turbinate size reduction. None of the participants experienced any major adverse effects such as bleeding, synechia formation, and rhinitis sicca during or after the procedure.

To investigate the coblation’s effect in children with inferior turbinate hypertrophy, a study was conducted on patients aged 6 to 18 years old [33]. The study found the subjective nasal obstruction achieved 100% improvement as the preoperative symptom score dropped significantly, from 9 (range, 7–10) to 0 postoperatively. Hence, this technique has been shown to be a safe and effective technique for the treatment of children with nasal obstruction, with none of the patients experiencing mucosal ulceration, adhesions, or other complications. The short-term beneficial effect of coblation at three months was further demonstrated subjectively and objectively by Farmer et al. [34], with significant improvement in nasal resistance assessed by rhinomanometry and the visual analogue scale for nasal obstruction. To determine the long-term effect of coblation, the same investigators [35] followed their patients for of up to 32 months. They observed sustained improvement in the subjective nasal obstruction and objective nasal resistance over the long term.

The weakness of coblation’s Reflex Ultra wand is its limited effect on the turbinate bone. To surmount this limitation, the Turbinator wand was developed and introduced. The Reflex Ultra is shaped as a fine-tipped wand and was designed to shrink the turbinate soft tissue using thermal energy. In contrast, the Turbinator has a wide tip with thermal effect similar to that of a Reflex Ultra but has a cutting effect comparable to that of a microdebrider. Thus, in addition to its thermal effect on the soft tissue, it has the ability to dissect the soft erectile tissue from turbinate bone. In a study by Mehta et al. [36], the Turbinator wand was compared with the Reflex Ultra wand. In one arm, the patients were subjected to turbinoplasty with Turbinator, and the other arm underwent Reflex Ultra. Visual analogue scale and endoscopic assessments were done postoperatively at day 7, first month, third month, and first year. Both groups reported significant and matching results in the long term, but an immediate improvement at one week was seen in the Turbinator group only. The significant difference in the outcome is accredited to the Turbinator’s cutting action on the submucosal tissue and bony tissue, leading to an instant reduction in the turbinate’s size and symptom improvement.

### 3.4. Radiofrequency Turbinoplasty

Radiofrequency turbinoplasty is a minimally invasive technique that reduces the turbinate volume in a precise and targeted manner. It uses radiofrequency to reduce the tissue volume with minimal impact on surrounding tissues [37]. Radiofrequency is mainly applied as an intraturbinoplasty method (Figure 5). The surgical steps are similar to those of coblation, except there is no saline necessary for its application. Li et al. [38] demonstrated radiofrequency-relieved nasal obstruction with a total mean reduction of 56.5% postoperatively and with minimal adverse effects. The use of radiofrequency involves energy in the range of 60 °C to 90 °C, which minimizes excessive tissue injury and limits heat dissipation. Other studies demonstrated identical effects of radiofrequency in their patients. The significant improvement of nasal obstruction was achieved in 85.5% of patients with none or mild postoperative pain [13,37,39,40,41]. There were only minimal adverse reactions reported, such as crusting, adhesion, dryness, or nasal bleeding. All studies reported significant nasal flow increase on rhinomanometric measurements, with an average of 55% decrease in the severity of nasal obstructive symptoms [13,41,42]. A comparative study between radiofrequency turbinoplasty and microdebrider turbinoplasty demonstrated a marked significant improvement in nasal flow with no significant difference either in visual analogue scale or in rhinomanometry score between these two techniques [43]. A study that assessed clinical outcomes of radiofrequency turbinoplasty against conventional turbinectomy reported good improvement of the nasal function and similar volume reduction of the inferior turbinates in both methods [44]. However, mucociliary function was observed to be affected more in the conventional turbinectomy. Interestingly, another study reported that treating patients who had both septal deviations and inferior turbinate hypertrophy by radiofrequency turbinoplasty alone had similar results with those who were treated by a combined radiofrequency turbinoplasty and septoplasty [45]. 

### 3.5. Ultrasound Turbinoplasty

Ultrasound technology for rhinologic surgery is a relatively newer technique. Gindros et al. [46] used the Lora Don 3 equipment (Diamant Co., Thessaloniki, Greece) to treat their patients with inferior turbinate hypertrophy. They performed it by inserting an activated ultrasonic nasal probe submucosally through the inferior turbinate and advancing it along its length. Then, the probe was moved in a forward–backward movement in a slow and gentle manner. At the end of this process, the nasal probe was withdrawn from the turbinate tissue. These steps can be repeated in extensive turbinate enlargement by creating one or two more parallel tunnels. Exposure of the affected tissues of inferior turbinate to submucous low-frequency fluctuation of an ultrasonic nasal probe resulted in destruction of the cavernous and connecting tissues, with subsequent reduction of the volume of the turbinate. Thus, fast restoration of nasal function and respiratory function could be obtained.

Gindros et al. [46] enrolled patients with medically refractory chronic nasal obstruction due to inferior turbinate enlargement; one group underwent inferior turbinate volume reduction using ultrasound procedure on the left side and monopolar diathermy on the right, and the other group underwent coblation technique on the left side and ultrasound turbinate reduction on the right. Most patients of all groups experienced great improvement of their symptoms. Visual analogue scale scores were better in the group treated by ultrasound in comparison with those treated by coblation and electrocautery. This improvement was further confirmed by objective testing, which showed the ultrasound turbinate reduction procedure had better results, with decreased nasal resistance, increased nasal flow, and significant increase in nasal patency.

## 4. Adjunct Technique

Out-fracture of the inferior turbinate may be performed with other turbinate reduction techniques. It involves lateral displacement of the inferior turbinate by an initial in-fracture, moving it medially from its attachment at the lateral nasal wall. The basis of doing this procedure is to create additional space when inferior turbinate is lateralized. Its efficacy is variable, resulting in much criticism over its role. As there is a tendency for the turbinate to return to its original position, it is not recommended as a single procedure, but it may be used to supplement the other techniques [13].

## 5. Differences and Similarities of Surgical Approach to Inferior Turbinate Hypertrophy between Children and Adults

When compared to adults, turbinate reduction surgery in children is contentious and debatable. The considerations are whether the benefits outweigh the risks, and medical therapy should always be optimized before surgical therapy is contemplated. The apprehension is of needless complications like excessive bleeding, damage to the mucosa with synechie and tear, disruption of nasal physiology and function, and disturbance of facial growth and development. Hence, any surgery of turbinates in childhood should be limited to avoid unwarranted complications. With this in mind, a conservative procedure involving turbinate reduction is ideal, whereby the turbinate mucosa is preserved, allowing the procedure to be accomplished without any harmful effect. 

Nonetheless, several studies carried out in children have not shown any significant harmful effects. Partial or total turbinectomies carried out in children aged between 9 and 15 years old achieved a success rate of 68% without any complications such as bleeding, synechia, or olfactory dysfunction [47]. A total turbinectomy performed in children less than 16 years was found to attain a success rate of 91% without any reported complications of crusting, atrophic rhinitis, or ozaena [48]. Another study of total turbinectomy in children less than 10 years old showed effectiveness approximately of 80%, but 6% of children developed synechiea formation [49]; no midfacial growth was noted with a follow- up period of over 14 years.

With the available newer technologies such as microdebrider or radiofrequency performed as an intraturbinoplasty technique, such procedures can be performed as mucosal-sparing methods with greater efficacy and minimal postoperative complications [50]. Remarkably, the mucosal-sparing approach has been shown to be safe and efficacious when performed concurrently with other procedures such as adenotonsillectomy [51]. Due to limited data and uncertainty over the long-term adverse effects, the decision to perform surgical turbinate reduction in children depends on clinical judgement and the preference of patients or their guardians/parents. In terms of surgical steps, there are no differences between children and adults.

## 6. Effects of Inferior Turbinate Surgery on Nasal Physiology and Function

In hypertrophied turbinate mucosa, histological examination revealed that there is an overall increase of the lamina propria width together with the thickening of mucosa overlying the medial portion of the turbinate, which accounted for the turbinate size increase [52]. In addition, there is also engorgement of the venous sinusoid within the hypertrophied turbinate. The connective tissue, submucosal glands, and vessels, however, remained relatively unchanged. Postoperatively, it is important to maintain the function of the mucociliary clearance by preserving the mucosal surface epithelium. Any procedure targeting a reduction in the size of the turbinates should be able to improve or preserve the mucosal morphology alongside improvement of the nasal airway. A reflection of these can be seen from the presence of normal epithelial goblet cells, lamina propria glands, vasculature, and tissue eosinophilia in the mucosal epithelium to indicate the restoration of epithelial function. One study reported reduction of submucosal glands and venous sinusoids, epithelial stripping, significant fibrosis, and pockets of squamous metaplasia in the operated areas [53]. In contrast, the epithelial structure showed normal nasal mucosa following surgery in another study, believed to be due to its regenerative capacity [54].

A study demonstrated that the cross-sectional area of the nasal cavity in patients with chronic nasal obstruction requiring septorhinoplasty procedure significantly changed when turbinate reduction surgery was added to the former procedure [55]. Acoustic rhinometry postoperatively revealed that the minimal cross-sectional area at the level of the nasal valve significantly increased in patients with the additional turbinate reduction surgery compared with those without. To determine the effect of inferior turbinate surgery on nasal physiology, a study was performed by Pelen et al. [56]. An acoustic rhinometry was performed on patients suffering from chronic nasal obstruction following two techniques; one group treated by radiofrequency and the other group treated by microdebrider. Both techniques were shown to be effective in relieving obstruction without any disruption of the nasal physiology.

## 7. What Is the Ideal Surgical Technique for the Reduction of Inferior Turbinate Hypertrophy?

Even though there is no gold standard in treating turbinate hypertrophy, in a best practice review [57], microdebrider turbinoplasty has been shown to be the most effective and safe. The review highlighted that ultrasound turbinoplasty is the next most promising technology and has the potential to be the leading surgical technique. Based on the present review, we support microdebrider turbinoplasty as one of the recommended techniques, but there is recent evidence that radiofrequency technology, particularly coblation, may also offer corresponding benefits [58]. Both have strong evidence from clinical trials and studies to demonstrate their efficacy and safety. Unfortunately, ultrasound turbinoplasty has never caught on, and there is scarcity of data to support its role in turbinate reduction. Moreover, both microdebrider turbinoplasty and radiofrequency technology have the advantage of widespread use and familiarity. On the other hand, there are other factors that should be considered in determining the outcome, such as criteria for patient selection. Turbinate hypertrophy consists of either mucosal or osseous hypertrophy or both, and a proper evaluation prior to surgery could assist in deciding the best technique to address the contributing components of the obstructed nose.

Nevertheless, there appears to be a significant detachment between current practices and the recommended surgical method, as can be seen by the wide availability of techniques being performed. It is noteworthy that the results of a survey of the “American Society for Aesthetic Plastic Surgery”, which includes practicing plastic surgeons and otorhinolaryngologists, showed 61.9% of respondents prefer conventional turbinoplasty, 35.2% prefer out-fracture of inferior turbinate (as a sole procedure), and only 8.6% give radiofrequency technology as their preferred surgical management strategy for inferior turbinate hypertrophy [59]. There appears to be disparity between clinical practice and medical evidences. It is concerning, as this may result in poor decision-making and selecting technique with inferior outcome. The conflict may be explained by confusion over the variety of available surgical techniques and lack of understanding of the rationale for performing the turbinate reduction. Besides providing a concise overview of each technique, hopefully this review can also serve as a guide to choose the optimal technique for turbinate reduction.

## 8. Conclusions

The main goals of surgical reduction of inferior turbinate hypertrophy are the relief of nasal obstruction and avoiding complications such as bleeding, crusting, and excessive pain. It is indicated and could be the treatment of choice when nasal block is refractory to medications. Although there is lack of consensus on the ideal methods, techniques such as microdebrider turbinoplasty and radiofrequency technology appear to have the advantage. Irrespectively, a judicious and cautious approach to turbinate resection is required to prevent complications. Which technique to perform may ultimately depend on the clinical practice, clinical skill, and experience of the surgeons.

## Figures and Tables

**Figure 1 ijerph-18-03441-f001:**
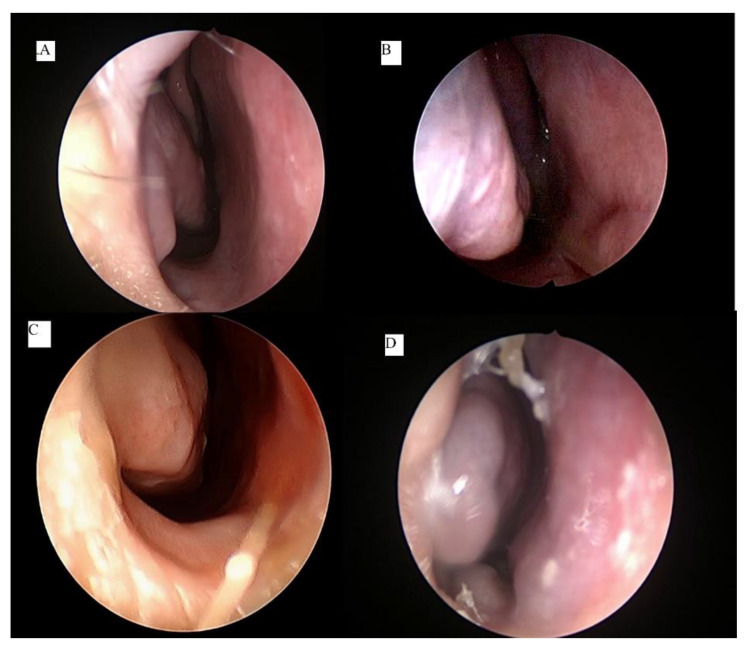
Nasoendoscopic grading system of inferior turbinate hypertrophy; grade 1 is 0–25% of total airway space occupied (**A**), grade 2 is 26–50% occupied (**B**), grade 3 is 51–75% occupied (**C**), and grade 4 is 76–100% occupied (**D**).

**Figure 2 ijerph-18-03441-f002:**
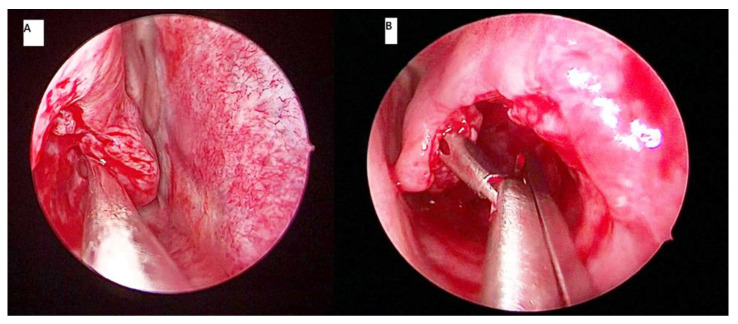
In conventional turbinoplasty, a freer elevator dissects the turbinate bone and soft erectile tissue (**A**) and a Blakeley’s forceps later removes part of the soft tissue and bone (**B**).

**Figure 3 ijerph-18-03441-f003:**
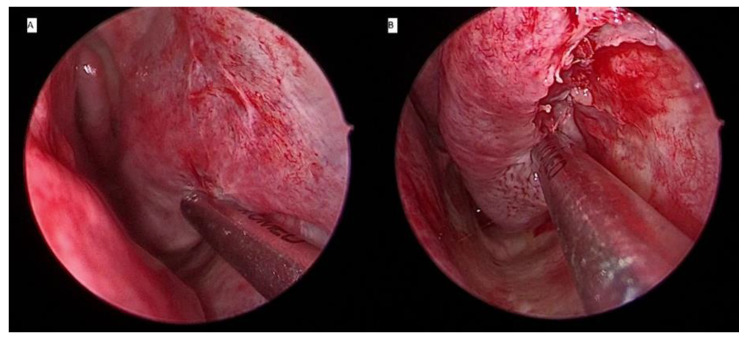
Microdebrider blade is placed at the lateral part of the left turbinate to dissect the lateral mucosal wall and turbinate bone from anterior (**A**) to posterior (**B**) direction.

**Figure 4 ijerph-18-03441-f004:**
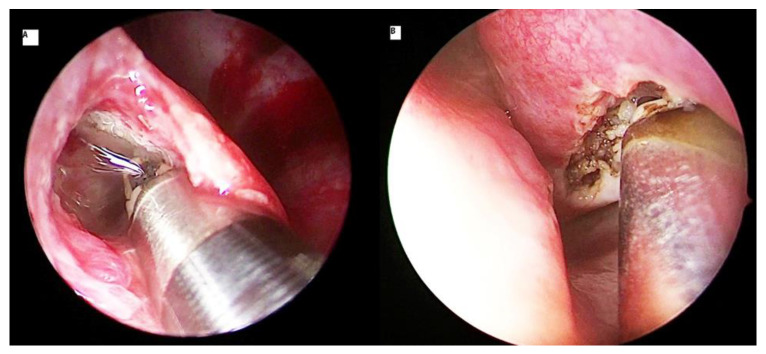
Coblation turbinoplasy applied as an intraturbinoplasty technique (**A**) and extraturbinoplasty technique (**B**).

**Figure 5 ijerph-18-03441-f005:**
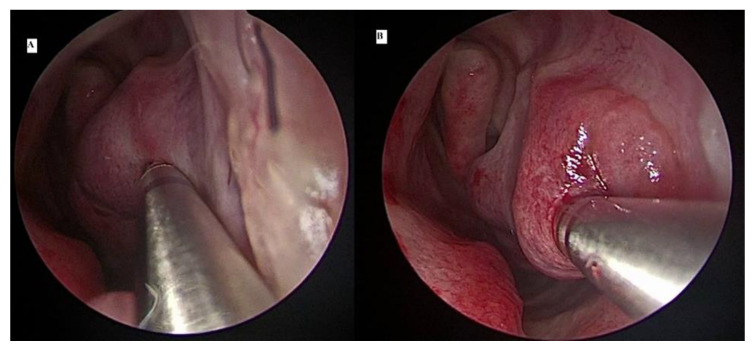
Radiofrequency probe is inserted at anterior head of the inferior turbinate and pushed in an anterior (**A**) to posterior (**B**) direction.

**Table 1 ijerph-18-03441-t001:** Type of surgical technique for turbinate reduction.

Nonmucosal Preservation Surgery	Mucosal Preservation Surgery
Conventional turbinectomy (partial or total)	Conventional turbinoplasty
Electrocautery turbinectomy	Microdebrider turbinoplasty
Laser turbinectomy	Coblation turbinoplasty
Cryoturbinectomy	Radiofrequency turbinoplasty
	Ultrasound turbinoplasty

## Data Availability

No new data were created or analyzed in this study. Data sharing is not applicable to this article.

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
