# Peer review of "Surgical Interventions for Inferior Turbinate Hypertrophy: A Comprehensive Review of Current Techniques and Technologies"

_ijerph, 2021, doi:10.3390/ijerph18073441_

Round 1

Reviewer 1 Report

The authors prepared the overview of midlle turbinate reduction techniques. The text is valid and content near all of surgical methods used for the turbinate )mucosal) reduction. 

The main troubleshoting point of this paper are:

a) absence of objective investigation of nasal obstruction. No any data about the rinometry, rinomano metry, mucosal impedance measrument. The classification of the obstruction is low. 

b) Abscence of indication of different approaches for any type of nasal obstruction by type or form of turbinates. What was the impact to the physiology of nasal cavity? 

c) total abscence of impact of different techniques to nasal function. 

I recommand the some corrections . 

Author Response

The authors prepared the overview of middle turbinate reduction techniques. The text is valid and content near all of surgical methods used for the turbinate mucosal reduction. 

The main troubleshooting point of this paper are:

Clarification 1

a) absence of objective investigation of nasal obstruction. No any data about the rhinometry, rhinomanometry, mucosal impedance measurement. The classification of the obstruction is low. 

Response:

This issue is addressed under the subheading of “Evaluation of the inferior turbinate hypertrophy for surgical intervention” by adding the following texts:

“Nasal obstruction owing to inferior turbinate hypertrophy can be measured objectively using an active anterior rhinomanometry, by measuring the differential pressure between anterior and posterior portions of the nose [10]. It is derived from the volume of airflow measured separately in both noses before and after the administration of a decongestant. By this way, the amount of mucosal or bony contribution towards the nasal obstruction can be estimated. Any increase in the nasal mucosal impedance caused by the nasal obstruction can be identified by this evaluation but it can’t localize the site of intranasal obstruction. For better accuracy, it should be applied concurrently with acoustic rhinometry. The reflection of acoustic signal by the respective nasal anatomical components forms the basis of acoustic rhinometry measurement [10]. By this means, all nasal sections volumes can be measured and allows a reliable topographic obstruction to be identified. Corresponding to the active anterior rhinomanometry, the measurement before and after decongestion enables it to differentiate mucosal pathology from other causes. This procedure has good reproducibility, quick and invasive which makes it advantageous for examination of nasal pathology in children.  It has become the first-line method to confirm the location and level of nasal obstruction, and universally used as an objective measurement of the effectiveness of medical and surgical treatments.”

Clarification 2

b) Absence of indication of different approaches for any type of nasal obstruction by type or form of turbinates. What was the impact to the physiology of nasal cavity? 

Response:

This issue is discussed under a newly created subheading “6.Effects of inferior turbinate surgery on nasal physiology and function”.

Clarification 3

c) total absence of impact of different techniques to nasal function. 

Response: 

This issue is discussed under a newly created subheading “6.Effects of inferior turbinate surgery on nasal physiology and function”.

Reviewer 2 Report

This study is reviewed surgical methods applied for inferior turbinate hypertrophy. As inferior hypertrophy is commonly encountered problem, reviewing various surgical techniques is important.

However, review papers about various surgical techniques of inferior turbinate hypertrophy have been previously published (ex, Otolaryngol Clin North Am . 2018 Oct;51(5):919-928.) What is the priority of this manuscript for publication compared to previously published paper? I suggest that authors should focus on some specific points of interior turbinoplasty and re-write manuscript rather than list fragments of surgical techniques (ex, differences of inferior turbinoplasty between children and adults).

-Nasal obstruction and treatment of inferior turbinate hypertrophy in children might be different between children and adults (ex, duration of medical treatment, adenoid hypertrophy should be evaluated in patients with nasal obstruction in children). Authors should describe the differences and similarities of surgical approach to inferior turbinate hypertrophy between children and adults.

-In anatomy part of inferior turbinate, shot summary of blood supply needs to be added (Anat Sci Int. 2021 Jan;96(1):13-19.)

-In part of ‘evaluation of the inferior turbinate hypertrophy for surgical intervention’, presence of septal deviation, unilateral hypertrophy of inferior turbinate, and allergic rhinitis should be evaluated.

-In part of ‘indications for inferior turbinate reduction surgery’, authors wrote when conservative management fails, surgical treatment is indicated. Then, what should be the duration for medical treatment before surgery?

-In part of ‘ Conventional turbinoplasty, Radiofrequency turbinoplasty,’  endoscopic surgical view need to be added, just like figure 2.

-In part of ‘What is the ideal surgical technique for the reduction of inferior turbinate hypertrophy?’, authors conclusion is too narrow. The outcome of inferior turbinoplasty depends on various factors, and it could not be simply concluded.

Author Response

This study is reviewed surgical methods applied for inferior turbinate hypertrophy. As inferior hypertrophy is commonly encountered problem, reviewing various surgical techniques is important.

Clarification 1

However, review papers about various surgical techniques of inferior turbinate hypertrophy have been previously published (ex, Otolaryngol Clin North Am . 2018 Oct;51(5):919-928.) What is the priority of this manuscript for publication compared to previously published paper?

Response:

Thank you for bringing up this fine point. In the preparation of the present review article we have referred to the mentioned article published in Otolaryngol Clin North Am 2018. Whilst it is a comprehensive overview of turbinoplasty, we feel it lacks the detailed description of each specific technique and the means that the surgical technology contributes to overcome the limitations of earlier procedures. By classifying the surgical techniques into 2 types;1) mucosal sparing and 2) non-mucosal sparing, we aim to outline the principle involved for each specific procedure. We feel this is crucial to help in aiding understanding and clear the confusion due to myriad of turbinate reduction procedures being performed currently.

To make this point clearer, we have renumbered the mucosal sparing technique separately from the non- mucosal sparing technique.

Clarification 2

I suggest that authors should focus on some specific points of interior turbinoplasty and re-write manuscript rather than list fragments of surgical techniques (ex, differences of inferior turbinoplasty between children and adults). Nasal obstruction and treatment of inferior turbinate hypertrophy in children might be different between children and adults (ex, duration of medical treatment, adenoid hypertrophy should be evaluated in patients with nasal obstruction in children). Authors should describe the differences and similarities of surgical approach to inferior turbinate hypertrophy between children and adults.

Response:

The reviewer has highlighted an excellent point. To address the differences between children and adults turbinoplasty, we have created an additional subheading specifically to cover this important topic “5.Differences and similarities of surgical approach to inferior turbinate hypertrophy between children and adults”.

Clarification 4

-In anatomy part of inferior turbinate, short summary of blood supply needs to be added (Anat Sci Int. 2021 Jan;96(1):13-19.)

Response:

A short summary of the anatomy of inferior turbinate is added as follows: “The inferior turbinate is supplied by inferior thyroid artery (ITA), a branch of the posterior lateral nasal artery [4]. The inferior thyroid artery enters inferior turbinate at its posterosuperior end where it divides into two or three branches. It courses through the inferior turbinate in a bony canal wrapped by a fascial coat binding the ITA and canal tightly. This relation is the main reason for the prolonged bleeding following turbinate surgery, as the fascial coat prevents ITA from contracting.”

Clarification 5

-In part of ‘evaluation of the inferior turbinate hypertrophy for surgical intervention’, presence of septal deviation, unilateral hypertrophy of inferior turbinate, and allergic rhinitis should be evaluated.

Response:

This is added: “It is imperative during the evaluation to exclude the presence of septal deviation, allergic rhinitis or unilateral inferior turbinate hypertrophy [9]. The concomitant conditions should be addressed simultaneously with the hypertrophied turbinate to achieve the best outcome.”

Clarification 6

-In part of ‘indications for inferior turbinate reduction surgery’, authors wrote when conservative management fails, surgical treatment is indicated. Then, what should be the duration for medical treatment before surgery?

Response:

This is added: “Generally, the consensus for the point of surgical intervention is at 3 months ensuing failure of medical therapy to resolve nasal obstruction as a result of the turbinate hypertrophy [11]. However, if there is a concomitant rhinosinusitis, the medical treatment could be extended up to 6 months.”

Clarification 7

-In part of ‘Conventional turbinoplasty, Radiofrequency turbinoplasty’ endoscopic surgical view needs to be added, just like figure 2.

Response:

Two figures are added. Figure 2 for conventional turbinoplasty and figure 4 for radiofrequency turbinoplasty. The previous Figure 2 is renumbered as Figure 3 and previous Figure 3 renumbered as Figure 5.

Clarification 8

-In part of ‘What is the ideal surgical technique for the reduction of inferior turbinate hypertrophy?’, authors conclusion is too narrow. The outcome of inferior turbinoplasty depends on various factors, and it could not be simply concluded.

Response:

We concur with this view. We change the relevant section to be more inclusive as follows: “On the other hand, there are other factors that should be considered in determining the outcome such as criteria for patient selection. Turbinate hypertrophy consists of either mucosal or osseous hypertrophy or both, and a proper evaluation prior to surgery could assist in deciding the best technique to address the contributing components of the obstructed nose.”

Round 2

Reviewer 2 Report

Authors tried to answer to each comment and it much improved the manuscript.